# SWE-Bench+: Enhanced Coding Benchmark for LLMs

## Abstract

Large Language Models (LLMs) in Software Engineering (SE) can offer assistance for coding. To facilitate a rigorous evaluation of LLMs in practical coding contexts, Carlos et al. introduced the *SWE-Bench* dataset, which comprises 2,294 real-world GitHub issues. The *SWE-Bench* dataset has quickly become the most popular benchmark for evaluating LLMs in software engineering. It has been adopted by leading companies such as OpenAI, Anthropic, Google, and Meta to assess the coding capabilities of their models. Despite its central role in measuring the state-of-the-art performance of newly released LLMs, a systematic evaluation of the quality of *SWE-Bench* is still lacking.

In this paper, we addressed this gap by presenting an empirical analysis of two variants of *SWE-Bench* dataset (i.e., *SWE-Bench Lite* and *SWE-Bench Verified*), both of which have been validated by developers to ensure quality. We conducted a manual screening of instances where the top three models in the SWE-Bench leaderboard (i.e., SWE-AGENT 1.0, OPENHANDS + CODEACT v2.1, and AUTOCODEROVER-V2.0) successfully resolved issues by comparing the model-generated patches with the actual pull requests.

Our analysis reveals some critical issues with the *SWE-Bench* dataset: 1) 60.83% of the successfully resolved issues involve "solution leakage", where the solutions were either directly provided or indirectly hinted at in the issue report or comments. 2) 47.93% of the resolved issues were incorrectly marked as resolved due to patches passing weak test cases, i.e., the tests were not sufficient to verify patch correctness; we refer to these insufficiently verified patches as "plausible patches". When we filtered out these problematic issues, the resolution rate of the three agents dropped from 42.1% to 21.8% on average on *SWE-Bench Lite* and from 51.7% to 25.9% on average for *SWE-Bench Verified*.

The critical issues in the current *SWE-Bench* dataset motivated us to create a more rigorous evaluation framework, *SWE-Bench+*, by addressing solution-leak risks and enhancing test suites for patch validation. Specifically, we introduce `SoluLeakDetector`, an LLM-based tool to filter issues with solution leaks, and `TestEnhancer`, an LLM-based approach to strengthen test suites and mitigate weak test problems. `SoluLeakDetector` achieves 80.45% accuracy in solution-leak detection. `TestEnhancer` improves patch validation and identifies plausible patches for 97.11% of weak-test issues, leading to average resolution rate drops of 27.00 percentage points on *SWE-Bench Lite* and 36.27 on *SWE-Bench Verified*. Although we focus on *SWE-Bench*, our approach can be readily extended to other Software Engineering benchmarking datasets to support their evolution.

## 1 Introduction

The *SWE-Bench* dataset was created to systematically evaluate the capabilities of large language models (LLMs) in resolving software issues Jimenez et al. (2024). Given an issue description, the task for the LLM is to modify the corresponding codebase to produce a correct resolution. Each issue instance includes a textual description, a pull request referencing the associated buggy repository, a set of test cases that can be used to verify model-generated patches, and a gold patch made by developers to fix this issue. The original *SWE-Bench* contains randomly picked 2,294 issues from

12 projects hosted in GitHub. To improve the dataset's rigor, two manually curated variants have recently been introduced: *SWE-Bench Lite*[1] , which focuses on 300 bug-fixing issues, and *SWE-Bench Verified*[2], which contains 500 carefully validated issues with clear descriptions and strong test cases.

Since its release in 2023, the *SWE-Bench* dataset has rapidly become the leading benchmark for evaluating LLMs in Software Engineering. It has been adopted by major companies such as OpenAI, Anthropic, Google, DeepSeek, and Meta to assess the coding capabilities of their models. Within just a year, hundreds of LLM-based agents have been developed and tested on this benchmark Chen et al. (2024); Zhang et al. (2024a); Xia et al. (2024); Yang et al. (2024b); Zhang et al. (2024c); Rosa et al. (2024); Zan et al. (2024); Yu et al. (2025); Gao et al. (2025); Lei et al. (2024); Team & et al. (2025); Antoniades et al. (2025) However, despite its central role in evaluating state-of-the-art LLMs and agents in the Software Engineering domain, a systematic assessment of the quality of the *SWE-Bench* dataset remains missing.

In this paper, we investigate the quality of the *SWE-Bench* dataset through two key contributions. First, we conduct an empirical study of state-of-the-art SE agents on *SWE-Bench Lite & Verified*. This study examines: (1) the quality of issue descriptions and discussions (which are often wrapped in the prompts to the SE agents), specifically whether they contain hints that may inadvertently leak information about how to fix the issues, thereby simplifying the task; and (2) the adequacy of the test cases of issues, evaluating whether they are sufficiently strong to reliably validate the correctness of generated patches. Second, we introduce *SWE-Bench+*, a framework designed to enhance the *SWE-Bench* dataset by filtering out solution-leaking issues and strengthening issues' test cases to better detect and exclude suspicious patches.

During the time of our empirical study, we selected three top-performing approaches from the *SWE-Bench* leaderboard: *SWE-Agent 1.0 (Claude 3.5 Sonnet), OpenHands + CodeAct v2.1 (claude-3-5-sonnet-20241022)*, and *AutoCodeRover-v2.0 (Claude-3.5-Sonnet-20241022)*.

First, we gathered the commonly resolved issues from the *SWE-Bench Verified & Lite* dataset across the three agents, yielding a total of 217 issues and 651 patches (217 × 3). We achieved this by filtering only the instances with evaluation logs that showed all tests passed. Second, we conducted our empirical study through two steps: 1) **Manual issue quality checking**: We manually examined the issue descriptions, which are included in the inputs, to determine whether they reveal knowledge related to how to fix an issue. 2) **Patch validation study**: We compared the gold patches with the model-generated patches by analyzing the modified files, changed lines, and code alterations in both versions, based on which, we further evaluated the resulting code behavior based on the underlying logic. We then examined the corresponding test cases of each issue to determine whether they adequately covered the changes in the gold patches and were sufficient to distinguish gold patches from model-generated patches.

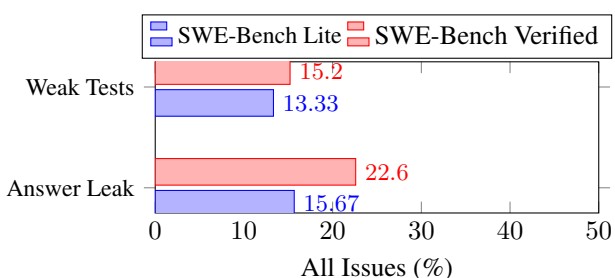

Figure 1: Distribution of problematic issues among all instances in *SWE-Bench Lite* (300 instances) and *SWE-Bench Verified* (500 instances)

Our empirical study identified five types of quality problems (see Table 1) in the issues in *SWE-Bench Verified & Lite* that significantly affect the reliability of LLM evaluation. We summarized them into two broadly types: 1) **Solution Leak:** In 60.83% of the resolved instances, the solutions to issues were either explicitly stated or subtly implied within the issue reports or comments. In other words, some issue descriptions directly include the solution code for the reported bug, while others offer hints or guidance on the general approach the solution should take. 2) **Weak Tests:** In 47.93% of the resolved instances, the changes made by the model are either incorrect, incomplete,

---

[1]https://www.swebench.com/lite.html

[2]https://openai.com/index/introducing-swe-bench-verified/

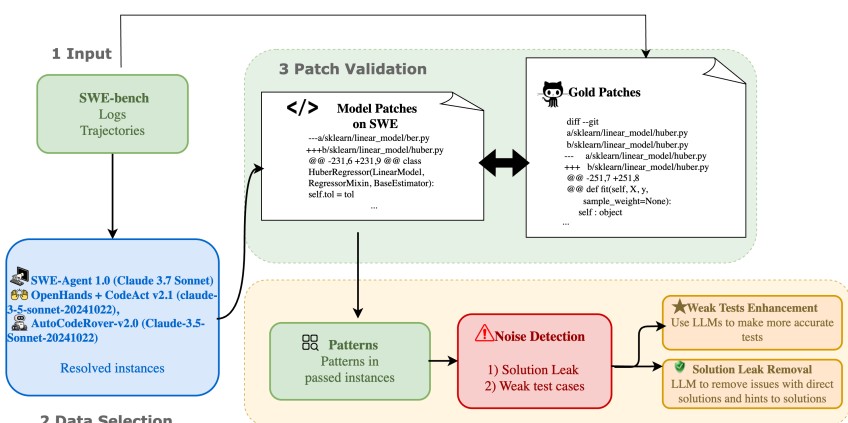

Figure 2: Overview of our analysis for *SWE-Bench* datasets

or applied to different files or functions compared to the gold patch. Despite these discrepancies, the changes pass the tests, indicating that the tests are too weak to catch such errors.

Figure 1 presents the distribution of issues in the two experimental datasets, i.e., *SWE-Bench Lite* (300 instances) and *SWE-Bench Verified* (500 instances). On *SWE-Bench Lite*, 15.67% of all its issues exhibit answer leakage problems, while on *SWE-Bench Verified*, the proportion is 22.6%, including both direct leaks and hint-based guidance. Additionally, 13.33% of all issues in *SWE-Bench Lite* and 15.2% in *SWE-Bench Verified* are identified as problematic due to weak test cases.

To address the quality issues in *SWE-Bench* datasets, we created a framework, i.e., *SWE-Bench+*, which targets 1) identifying the issues that include solutions in the issue description or comments and 2) enhancing the test suites for filtering out suspicious patches. Specifically, we introduce an LLM-based solution leak detection tool, i.e., `SoluLeakDetector`, to filter out issues with the solution leak problem, and an LLM-based test generation approach, i.e., `TestEnhancer`, to enhance the test suite of issues to avoid the weak test problem. Our evaluation shows that *SWE-Bench+* can achieve 80.45% accuracy on solution-leak-issue identification. `TestEnhancer` helps enhance the patch validation process and identify plausible patches for 97.11% issues that have weak tests, resulting in resolution rates dropping by an average of 27.00 percentage points on *SWE-Bench Lite* and 36.27 percentage points on *SWE-Bench Verified* across the three top-performing models. In addition, although we focus on *SWE-Bench*, our approach (i.e., solution-leak-issue identification and `TestEnhancer`) can be easily extended to other Software Engineering benchmarking datasets to support their evolution.

## 2 ISSUE QUALITY ANALYSIS OF *SWE-Bench*

We conducted an empirical study on the issues in the *SWE-Bench Lite* and *SWE-Bench Verified* datasets that can be resolved by the three agents (i.e., SWE-AGENT 1.0, OPENHANDS + CODE-ACT V2.1, and AUTOCODEROVER-V2.0). The goal of this study was to determine whether the issues exhibit potential quality problems that could affect the evaluation of LLMs. Figure 2 outlines the major steps we followed in our study. The input is the set of all involved issues in *SWE-Bench Lite* and *SWE-Bench Verified*. Each issue contains a description and the patch created by developers to address the issue, which we call a "gold patch". We selected three agents and refer to the patches they generated for each issue as "model-generated patches". We first examined the issue descriptions, user discussions, and other natural language information included in the prompt that guided the LLMs in producing a patch. We then compared the gold and model-generated patches to the issue by analyzing the corresponding files that were changed. As we studied the model-generated patches, we also examined the logs and trajectories generated by the model. Logs provide the step-by-step execution of the models. The trajectory data provide a detailed record of the models' decision-making processes while making a resolution as a patch. To reduce potential biases during

Table 1: Issue quality problems found among the 217 common fixed issues by the three models.

| Quality problems | Numbers (percentage) | Root cause |
|---|---|---|
| Solution leak | 70 (32.26%) | solution leakage |
| Solution hint leak | 62 (28.57%) | solution leakage |
| Incorrect fixes | 43 (19.82%) | weak tests |
| Different files/functions changed | 22 (10.14%) | weak tests |
| Incomplete fixes | 39 (17.97%) | weak tests |

the comparison between gold and model-generated patches, three authors independently performed the patch validation study. Each author carefully examined the files and lines changed, reviewed the issue descriptions, and evaluated the implementation styles and intentions behind both the model-generated and developer-generated patches. The disagreements were resolved through a broader discussion involving all the authors.

## 2.1 Quality Deficiencies in *SWE-Bench* Issues

We analyzed 217 commonly resolved issues across the three models, yielding 651 ($217 * 3$) patches. An issue is considered problematic if it meets either of two mutually exclusive conditions: (a) *Solution-leak (direct or hint)*, affecting 132/217 (60.83%), or (b) *Weak-test-only*, where at least one patch is incorrect, incomplete, or touches different files/functions among the remaining 85 issues, identifying 32/217 weak-test-only cases. Combined, 77.88% of commonly resolved issues are problematic. We categorize these into five patterns, i.e., two solution-leak-related and three weak-test-related, summarized in Table 1, with definitions, counts, and root causes discussed below.

**1. Solution leak:** represents instances where the solution to the issue is clearly outlined in the issue description or comments on GitHub. Since both the issue descriptions and comments (referred to as *hints_text* in the *SWE-Bench* study) are provided as input to the LLM-based issue resolution agents, these agents can extract the solutions directly from this information instead of generating them independently. 32.26% of the successfully resolved issues followed this pattern, making it the most common among resolved patches. This raises significant concerns about a model's actual performance and the validity of the SWE-Bench instances as benchmarks. If a model is simply copying the solution it already has access to, it isn't demonstrating true problem-solving capabilities but rather replicating what is provided, thus limiting the assessment of its ability to generate new solutions. The example shown in Figure 3a and Figure 8 (in the Appendix A.1) illustrates issue report 16669[3] from the *sympy* project, where the issue description provided the exact solution code patch required to resolve the issue, which makes it possible for the model to directly copy the solution from the issue report and generate the same solution as provided.

**2. Solution Hint Leak:** emerges when the descriptions or comments of issues contain partial information or indirect suggestions that guide models toward the solution without explicitly providing the complete fix. This pattern was present in 28.57% of issues. For instance, as shown in Figure 3b, a solution hint is provided in the problem description of the issue. While it does not provide the complete implementation, it gives a clear direction about which function should be used and where it should be applied. Such hints can influence how a model approaches the solution.

**3. Incorrect fixes:** refer to cases where the model-generated patches provide incorrect solutions, yet pass the test cases when they should have failed. This pattern was present in 19.28% of the passed instances, suggesting a weakness in test cases where the functionality of the issue resolution is not correctly captured. The fact that incorrect patches can pass the test cases raises suspicion about the relevance and accuracy of the test cases in assessing whether the issue has been fully resolved. Figure 9 (in the Appendix A.1) shows a comparison between the model-generated patch and the gold patch for django-32517[4]. According to the issue description, a new functionality is needed to reverse a Python *OrderedSet* by implementing the *__reversed__* function. The gold patch demonstrates the correct behavior, where the entire dictionary is reversed, while the generated patch only reverses the dictionary's keys. As a result, the two patches produce entirely different outputs, as they apply different methods to the dictionary.

---

[3] https://github.com/sympy/sympy/issues/16669
[4] https://code.djangoproject.com/ticket/32517

```
description of the issue:

The SQLCompiler is incorrectly removing "
    order_by" clauses because it
    determines the clause was already "
    seen" in SQLCompiler.get\_order\_by()
    . The issue occurs with expressions
    written as multiline RawSQL.The bug
    is located in SQLCompiler.
    get_order_by(),  specifically at:
    without_ordering = self.
    ordering_parts.search(sql).group(1)

As a quick/temporal fix I can suggest
    making sql variable clean of newline
    characters, like this:
sql_oneline = '␣'.join(sql.split('\n'))
without_ordering = self.ordering_parts.
    search(sql_oneline).group(1)
```

```
description of the issue:

A security vulnerability exists in the
    password reset functionality where
    tokens remain valid even after a user
     changes their email address. The
    current implementation of
    PasswordResetTokenGenerator does not
    consider email address changes when
    validating tokens.

The fix is to add the user email address
    into PasswordResetTokenGenerator.
    _make_hash_value()
Nothing forces a user to even have an
    email as per AbstractBaseUser.
    Perhaps the token generation method
    could be factored out onto the model,
     ala get_session_auth_hash().
```

(a) A *Direct Solution Leak* example (django-11001)  (b) An example of *Hint Leak* (django-13551)

Figure 3: Side-by-side examples of (a) direct solution leak and (b) hint leak in problem statements.

**4. Different files/functions changed:** This pattern refers to cases where the model-generated patches modify files or functions unrelated to the issue at hand. These files differ from those altered in the gold patch, yet the model's patches still pass the test cases despite this discrepancy. This pattern appears in 10.14% of the passed instances. This pattern highlights a weakness in the model's ability to accurately locate and address the source of the issue. The fact that the test cases pass, even though changes were made in irrelevant files, suggests that the test cases are either weak or irrelevant and should have failed in detecting the incorrect modifications. Figure 10 (in the Appendix A.1) presents an example from issue-26093 of the Matplotlib project[5], where the model-generated patch modifies the *cbook.py* file, while the gold patch makes changes to the *_axes.py* file. This shows that the model's patch affects a completely different file from the gold patch, highlighting the model's inability to accurately identify the correct file containing the bug.

**5. Incomplete fixes:** This pattern refers to model-generated patches that offer incomplete implementations compared to the gold patches, often omitting critical details. This pattern appears in 17.97% of passed instances. For instance, some patches include only partial if-else statements, neglecting edge cases that the gold patch addresses. Although the model-generated patches follow the correct implementation approach, they overlook important aspects that could lead to failures in production or when handling edge cases. This underscores a weakness in the test cases, as they fail to capture the finer details necessary for a comprehensive issue resolution.

The example provided in Figure 11 (in the Appendix A.1) shows the same change being made by the model and the one made by the developers in the gold patch[6]. The gold patch provides a complete fix, while the model patch provides a partial fix. Specifically, the gold patch properly handles the detection of an event loop in the current thread by including a *try-except* block to catch *RuntimeError* when an event loop is unavailable and checks if the event loop is running before raising an exception. Additionally, it wraps the entire logic in a condition that checks the environment variable *DJANGO_ALLOW_ASYNC_UNSAFE*. In contrast, the generated patch is missing critical parts of this logic, such as the *try-except* block and the check for a running event loop. As a result, the model-generated patch is incomplete, missing key error handling and flow control that are necessary for ensuring safe operation.

## 2.2 IMPACT OF SOLUTION (HINT) LEAK

To quantitatively assess the impact of solution leakage on model performance, we conducted an empirical study using the three models evaluated in this paper. Among the 217 commonly resolved instances, we identified 132 instances (60.83%) containing some form of solution leakage, with 70 instances exhibiting direct solution leaks and 62 instances containing solution hint leaks. Before

---

[5]https://github.com/matplotlib/matplotlib/issues/26093
[6]https://code.djangoproject.com/ticket/31056

leakage removal, all 132 instances were successfully resolved by the models. We then automatically removed all solution-related information, including direct code solutions, suggestions, and hints (e.g., expected behavior) from the issue descriptions and re-evaluated the models' performance to determine if they could still fix the issues without leaked information.

Figure 4 shows that resolution rates dropped substantially after removing leaked information. OpenHands+CodeAct-v2.1 resolved 52/132 instances (39.39%, a 60.61-point drop), AutoCodeRover-v2.0 resolved 87/132 (65.91%), and SWE-Agent-v1.0 resolved 70/132 (53.03%), demonstrating the positive impact of solution leaks. Examining leakage types, for 70 solution-leak-direct instances, AutoCodeRover-v2.0 led with 70% (49/70), SWE-Agent-1.0 65.71% (46/70), and OpenHands+CodeAct-v2.1 35.71% (25/70). For 62 solution-leak-hint instances, AutoCodeRover-v2.0 achieved 61.29% (38/62), OpenHands+CodeAct-v2.1 43.55% (27/62), and SWE-Agent-1.0 38.71% (24/62). Higher resolution rates for direct-solution issues indicate that direct solutions were particularly helpful for the models.

## 2.3 IMPACT OF WEAK TESTS

Given that 77.88% of commonly resolved issues are problematic, we quantify how weak-test-only cases inflate headline resolution rates. For each model and dataset (i.e., *SWE-Bench Lite* and *SWE-Bench Verified*), we recompute resolution rates after excluding only weak-test-only issues, keeping solution-leak cases intact to isolate the impact of weak tests. Following the weak-test breakdown in Table 1, Table 2 reports the distribution of weak-test patterns among passed patches (Lite: n=82, Verified: n=179). Using the protocol from Section 2.1, we manually review each passed patch,

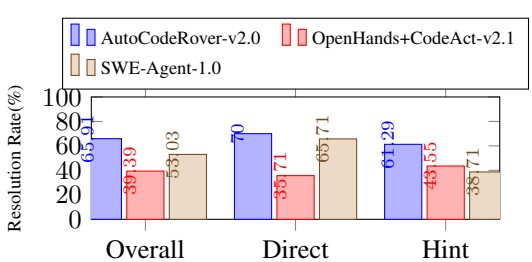

Figure 4: Resolution rates after leakage removal

flag weak-test-only cases, and recount them as failures while keeping the instance set and denominator unchanged, yielding the weak-test-removed resolution rate.

In *SWE-Bench Lite*, *AutoCodeRover-v2.0* produced incorrect fixes in 10.98% of cases, incomplete fixes in 25.61%, and fixes involving different files or functions in 3.66%. *OpenHands+CodeAct-v2.1* showed incorrect fixes in 9.76% of cases, incomplete fixes in 17.07%, and different files/functions in 8.54%. *SWE-Agent-v1.0* generated incorrect fixes in 10.98% of cases, incomplete fixes in 15.85%, and different files/functions in 6.10%. In *SWE-Bench Verified*, *AutoCodeRover-v2.0* produced incorrect fixes in 12.29% of cases, incomplete fixes in 10.61%, and fixes involving different files/functions in 5.03%. *OpenHands+CodeAct-v2.1* showed incorrect fixes in 11.17% of cases, incomplete fixes in 10.61%, and different files/functions in 5.59%. *SWE-Agent-v1.0* generated incorrect fixes in 13.97% of cases, incomplete fixes in 8.94%, and different files/functions in 3.91%. These suspicious fix patterns resulted in a substantial reduction in the actual resolution rates across all models.

Figure 5 contrasts the original resolution rates with weak test issues removed: we treat those instances with weak test patterns as failures, leaving the instance set and denominator unchanged. The difference shows how weak tests inflated the original rates. The new resolution criteria resulted in significant drops in the resolution percentages across all models. For *AutoCodeRover-v2.0 (Claude-3.5-Sonnet-20241022)*, the performance drops by 21.33 percentage points (from 37.33% to 16%) on *SWE-Bench Lite* and drops by 26 percentage points from 45% to 19% on *SWE-Bench Verified*. *OpenHands + CodeAct v2.1* showed a decline from 42% to 22% on *Lite* and from 52.4% to 26.8% on *Verified*. *SWE-agent 1.0 (Claude 3.5)* experienced a drop from 47% to 27.33% on *Lite* and from 57.6% to 31.8% on *Verified* when considering only correct fixes.

## 3 BUILDING SWE-BENCH+

To address the issues of the current *SWE-Bench* datasets and ensure a more accurate evaluation of the models' effectiveness in resolving issues, we create a more rigorous evaluation dataset *SWE-Bench+* by filtering out suspicious patches in *SWE-Bench*. Specifically, we introduce an LLM-based

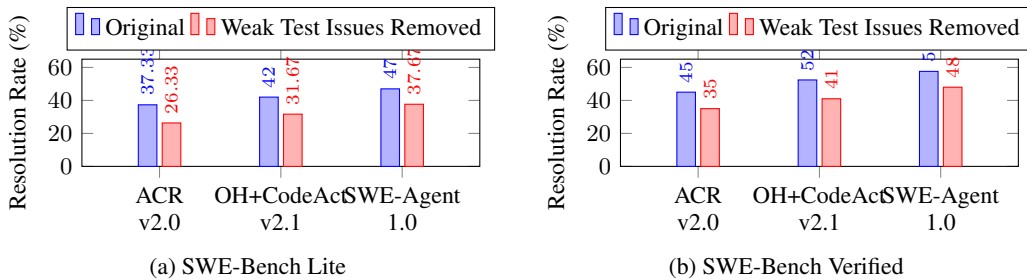

Figure 5: Resolution rate changes (Original vs Weak Test Removed) for three models.

Table 2: Patches Passing Due to Weak Test Cases (aka. plausible patches): *SWE-Bench Lite* (82 out of 300) vs. *SWE-Bench Verified* (179 out of 500 )

| SWE-Bench Lite (n=82) | | | |
|---|---|---|---|
| **Weak tests** | **AutoCodeRover-v2.0** | **OpenHands+CodeAct-v2.1** | **SWE-Agent-v1.0** |
| Incorrect fixes | 9 (10.98%) | 8 (9.76%) | 9 (10.98%) |
| Incomplete fixes | 21 (25.61%) | 14 (17.07%) | 13 (15.85%) |
| Different files/functions | 3 (3.66%) | 7 (8.54%) | 5 (6.10%) |
| SWE-Bench Verified (n=179) | | | |
| **Weak tests** | **AutoCodeRover-v2.0** | **OpenHands+CodeAct-v2.1** | **SWE-Agent-v1.0** |
| Incorrect fixes | 22 (12.29%) | 20 (11.17%) | 25 (13.97%) |
| Incomplete fixes | 19 (10.61%) | 19 (10.61%) | 16 (8.94%) |
| Different files/functions | 9 (5.03%) | 10 (5.59%) | 7 (3.91%) |

solution leak detection tool, i.e., `SoluLeakDetector`, to filter out issues with the solution leak problem, and an LLM-based test generation approach, i.e., `TestEnhancer`, to enhance the test suite of issues to avoid the weak test problem.

### 3.1 SOLULEAKDETECTOR: LLM-BASED SOLUTION LEAK ISSUE DETECTION

Our study revealed the persistent presence of solution leakage in *SWE-Bench* instances, where issue reports often contain direct or indirect hints about their resolution. To address this, we developed `SoluLeakDetector`, an LLM-based technique designed to systematically identify and filter instances in *SWE-Bench* that exhibit either direct solution leaks or solution hint leaks.

`SoluLeakDetector` leverages *GPT-4* to categorize instances into three distinct groups: (1) instances containing direct solution leaks, (2) instances with hint-based solution leaks, and (3) instances free from any form of solution leakage. The classification is performed using a three-shot prompting technique, where three clear examples of each category are provided to guide *GPT-4* in categorizing *SWE-Bench* instances accordingly. The model is prompted with issue descriptions extracted from the root of the instance's pull request. Additionally, `SoluLeakDetector` identifies and extracts potential hints or direct solutions from the issue description and generates an explanation detailing why the extracted fragment is classified as either a hint or a direct solution. We show the prompts in Appendix A.3.2.

### 3.2 TESTENHANCER: LLM-BASED TESTS ENHANCEMENT

Our study highlights that 47.93% of issues have plausible patches described in Table 1 are due to weak tests. To address this issue, we develop `TestEnhancer`, an automated test generation and validation process aimed at enhancing the original *SWE-Bench* test patches.

Our workflow is outlined in Figure 6. We begin by selecting specific attributes from the *SWE-Bench* dataset. Specifically, we extract the following attributes: *instance_id*, *gold_patches*, *test_patches*, and *base_commits*. The *base_commits* represent the commit IDs leading to the buggy version of each instance before the corresponding pull request (PR). To retrieve the buggy code patches, we trace these *base_commits* and extract the affected source code files from the repository. Next, we

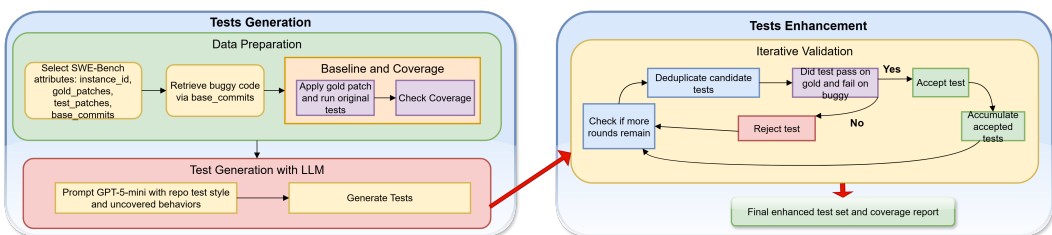

Figure 6: Test enhancement workflow

identify the specific files modified in the *gold patch* and apply *REGEX-based pattern matching* to extract the functions or methods that were changed. We show the prompts in Appendix A.3.1.

After applying the gold patch and running the original tests, `TestEnhancer` analyzes coverage gaps in modified files and prompts GPT-5-mini to generate enhanced tests that follow repository conventions, exercise uncovered behavior, and include regression tests failing on the buggy version but passing on the gold patch. Iteratively, it generates candidate tests, deduplicates overlaps, and runs them in isolated Docker containers to retain only those that meet the fail-to-pass criterion. Across rounds, it accumulates a curated set of tests that maximize behavioral coverage.

# 4 EVALUATION OF *SWE-Bench+*

## 4.1 PERFORMANCE ANALYSIS OF SOLULEAKDETECTOR

To ensure the reliability of the classification, two of the authors independently reviewed the categorized instances by reviewing all issue reports for the instances in *SWE-Bench Lite* and *SWE-Bench Verified* for cases of solution leakage. This human evaluation process helped mitigate potential biases and verify the correctness of `SoluLeakDetector`'s predictions. The manual review confirmed that `SoluLeakDetector` achieved an overall 80.45% accuracy in correctly classifying solution leaks within *SWE-Bench Lite & Verified*. For each dataset, `SoluLeakDetector` achieves 85.11% on *SWE-Bench Lite* and 77.88% in *SWE-Bench Verified* as shown in Table 3.

After categorizing all instances, we constructed *SWE-Bench+*, a refined benchmark that excludes any instance exhibiting solution leakage. *SWE-Bench+* consists of 707 instances, derived from *SWE-Bench Lite & Verified,* that were verified to contain no direct or indirect solution hints, ensuring a more reliable evaluation dataset. By eliminating biased instances, *SWE-Bench+* provides a fairer and more robust framework for evaluating LLM-based models in real-world GitHub issue resolution.

Table 3: The accuracy of `SoluLeakDetector` on solution-leak detection.

| Dataset | # Solution-Leaks | Correctly Detected | Accuracy (%) |
|---|---|---|---|
| *SWE-Bench Lite* | 47 | 40 | 85.11 |
| *SWE-Bench Verified* | 113 | 88 | 77.88 |

## 4.2 PERFORMANCE ANALYSIS OF TESTENHANCER

To evaluate the performance of `TestEnhancer`, we applied the enhanced test suite to patches generated by the three models for issues in *SWE-bench Lite & Verified* and measured the reduction in resolution rate when applying the newly generated test cases.

Our evaluation revealed that `TestEnhancer` helps enhance the patch validation process and

Table 4: Resolution rates before and after the enhanced test suite (values in %). $\Delta$ is the percentage-point drop.

| Model | Dataset | Original | TE | $\Delta$ (pp) |
|---|---|---|---|---|
| AutoCodeRover-v2.0 | Lite | 37.33 | 10.00 | 27.33 |
| | Verified | 48.00 | 9.20 | 38.80 |
| OpenHands+CodeAct-v2.1 | Lite | 42.00 | 15.67 | 26.33 |
| | Verified | 52.40 | 18.20 | 34.20 |
| SWE-Agent-v1.0 | Lite | 47.00 | 19.67 | 27.33 |
| | Verified | 57.60 | 21.80 | 35.80 |

identify plausible patches for 97.11% of issues that have the weak tests problem. The enhanced test cases lead to significant drops in resolution rate across all models when evaluated with the enhanced test suite, as shown in Table 4. Our evaluation revealed substantial drops in resolution rates across all models when using the enhanced test suite. On *SWE-Bench Lite*, AutoCodeRover-v2.0 dropped from 37.33% (112 instances) to 10% (30 instances, 27.33-point drop), OpenHands+CodeAct-v2.1 from 42% (126) to 15.67% (47, 26.33-point drop), and SWE-Agent-v1.0 from 47% (141) to 19.67% (59, 27.33-point drop). On *SWE-Bench Verified*, AutoCodeRover-v2.0 fell from 48% (225) to 9.2% (46, 38.8-point drop), OpenHands+CodeAct-v2.1 from 52.4% (262) to 18.2% (91, 34.2-point drop), and SWE-Agent-v1.0 from 57.6% (288) to 21.8% (111, 35.8-point drop). Notably, `TestEnhancer` substantially improved coverage, increasing average line coverage from 33.39% to 55.55% (22.16-point gain). These results indicate that `TestEnhancer` effectively identifies plausible patches. The enhanced tests reveal flaws undetected by the original *SWE-Bench* suite, confirming that weak tests inflate apparent LLM patch success and demonstrating `TestEnhancer`'s utility as a more robust evaluation framework for LLM-generated patches.

## 5 RELATED WORK

**LLM for Software Engineering.** Large Language Models (LLMs) have emerged as powerful tools and demonstrated impressive capabilities in various software engineering tasks, including code generation Jiang et al. (2024); Li & Döhmen (2024); Chen et al. (2021); Luo et al. (2024); Du et al. (2024), program repair Zhang et al. (2024b); Yang et al. (2024a); de Fitero-Dominguez et al. (2024) and bug detection Alrashedy & Binjahlan (2024); Hossain et al. (2024). The development of code generation benchmarks has been crucial for evaluating LLM performance. Notably, HumanEval Chen et al. (2021) was introduced to assess the functional correctness of code generated by LLMs. Building on this foundation, AlphaCode Li et al. (2022) demonstrated competitive performance in solving complex programming problems. To address limitations in existing benchmarks, EvalPlus Liu et al. (2024) enhanced HumanEval with more comprehensive test cases and revealed a significant overestimation of LLM performance in previous evaluations. LLMs also have shown promising results in program repair and bug detection. For example, AlphaRepair Xia & Zhang (2022) employed a zero-shot learning approach that outperformed traditional automated program repair (APR) tools. Further research demonstrated that LLMs could surpass existing APR techniques, particularly when fine-tuned on domain-specific data Xia et al. (2023). The application of LLMs in bug detection with innovative approaches like FuzzGPT Deng et al. (2023b) and TitanFuzz Deng et al. (2023a), leveraging these models to generate edge-case test inputs and perform mutation-based fuzzing for deep learning libraries. Several comprehensive studies have explored LLM applications across various software engineering domains Fan et al. (2023); Hou et al. (2024), delved into the natural language to code generation Zan et al. (2023), and analyzed the evolution and performance of Code LLMs across different tasks Zheng et al. (2024).

## 6 CONCLUSION

In this paper, we present the first empirical study on the robustness of the *SWE-Bench* dataset. Our analysis reveals significant limitations in the original *SWE-Bench*, particularly solution leakage and weak test cases, which compromise the reliability of prior model evaluations. To address these challenges, we construct *SWE-Bench+*, a more rigorous evaluation dataset that filters out suspicious patches from *SWE-Bench*. Specifically, we introduce an LLM-based solution leak detection tool, `SoluLeakDetector`, to identify and remove issues affected by solution leakage. Additionally, we develop an LLM-based test generation approach, `TestEnhancer`, to strengthen test suites and mitigate weak test issues. Our evaluation shows that `SoluLeakDetector` can achieve 80.45% accuracy on solution-leak-issue identification and reveals the extent of weak test issues: `TestEnhancer` causes resolution rates to drop by an average of 27.00 percentage points on *SWE-Bench Lite* and 36.27 percentage points on *SWE-Bench Verified* across the three top-performing models. Furthermore, *SWE-Bench+* supports the continuous, automated evolution of the *SWE-Bench* dataset and can be applied to other software engineering benchmarks, promoting more reliable and robust future evaluations.

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

# A    APPENDIX

Table of Contents:

## A.1    APPENDIX A

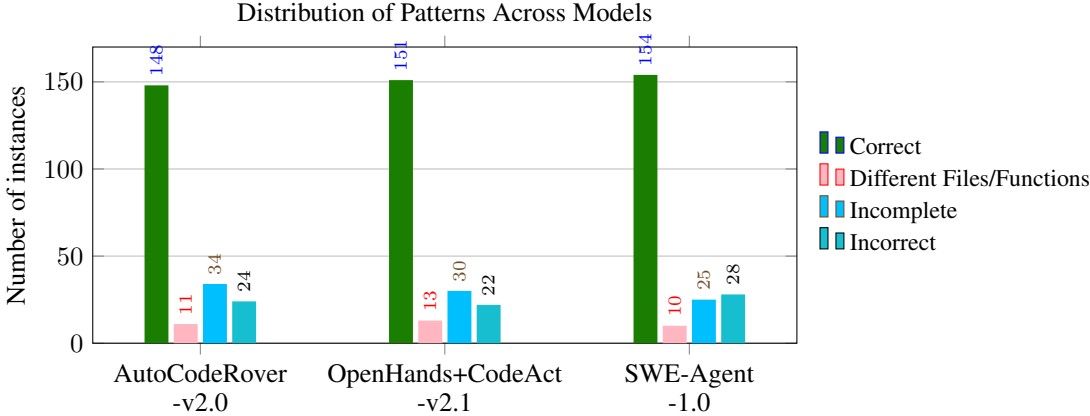

Figure 7: Counts per category for AutoCodeRover-v2.0, OpenHands+CodeAct-v2.1, and SWE-Agent-1.0.

## A.2    APPENDIX B

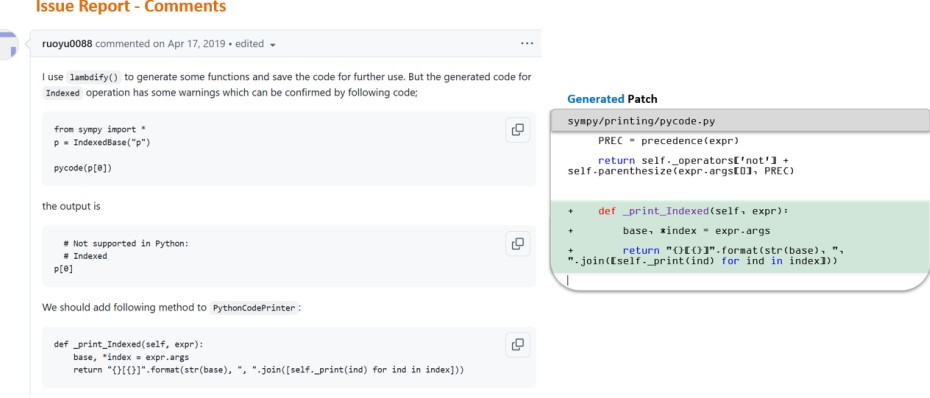

Figure 8: Solution Leakage in issue report for sympy-16669

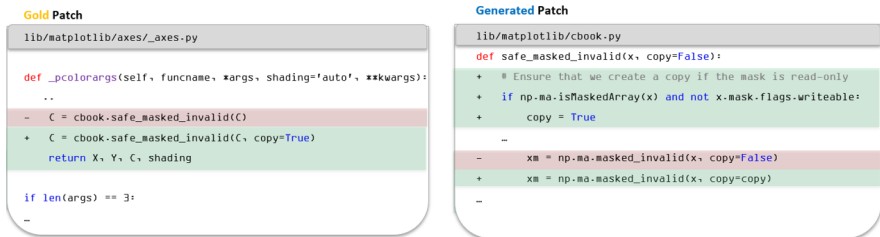

Figure 9: Incorrect fix generated by the model for xrray-7732

Figure 10: Different files changed by model for issue-26093 of Matplotlib

Figure 11: Incomplete fix generated by the model for django-31056

## A.3 APPENDIX C

### A.3.1 TESTENHANCER PROMPT

**You are an expert software engineer code assistant tasked with generating additional unit tests for a Python source file and its corresponding test file.**
**TASK:** Your objective is to add **30** regression tests that detect the bug, which is resolved by the provided patch.
**GUIDELINES:**

1. **Analyze the Code:** Examine the provided source code to understand its functionality, inputs, outputs, and core logic.

2. **Analyze the Patch:** Examine the provided patch to understand the bug it fixes in the source code.

3. **Identify Test Cases:** Develop a detailed list of test cases that will fully validate the provided patch.

4. **Add and Review Tests:** Integrate individual tests, ensuring they collectively cover all possible scenarios, including edge cases and exception handling.

5. **Maintain Consistency:** Ensure new tests are consistent with the existing test suite in terms of style, naming conventions, and structure. Assume new tests are part of the same suite if a test suite exists.

**ADDITIONAL CONSTRAINTS:**

- Return only valid Python in the YAML fields; do not produce syntax errors.
- Octal literals must use digits 0-7 only (e.g., 0o644). Do not generate invalid octal values like 0o800.

**Source File:** Here is the source file `source_file` that you will be writing tests against. We have manually added line numbers to assist in understanding the code coverage. These line numbers are not part of the original code.
`source_numbered`
**Patch:** Here is the patch that is applied to the source file `source_file` to fix a bug.
`patch_content`
**Test File:** Here is the file `test_file` that contains the existing tests.
`test_content`

**OUTPUT FORMAT:** The response should be only a valid YAML object, without any introduction text or follow-up text.
Example output:

```yaml
language: Python
number_of_tests: ...
test_behavior: ...
test_code: ...
new_imports_code: ...
```

### A.3.2   SoluLeakDetector Prompt

**You are a solution leakage detection expert.**

TASK

Analyze GitHub issue descriptions (`problem_statement`) and related comments (`hints_text`) for solution leakage.

DEFINITION

Solution leakage occurs when:

1. The solution is **explicitly mentioned**, such as:
   - Code snippets providing a direct fix.
   - Step-by-step instructions leading directly to the solution.
2. The solution is **subtly implied**, for example:
   - Explanatory text that indirectly leads to the correct fix.
   - Comments hinting at how to resolve the issue.

EXAMPLE 1

**Description:** I propose to add the following settings, with the following default values:

```
1 LANGUAGE_COOKIE_SECURE = False
2 LANGUAGE_COOKIE_HTTPONLY = False
3 LANGUAGE_COOKIE_SAMESITE = None
```

The default values maintain the current behavior.
These settings do not provide much security value, since the language is not secret or sensitive. This was also discussed briefly here: `https://github.com/django/django/pull/8380#discussion_r112448195`.
**Reasons for Change:**

- Sometimes auditors require them.
- I personally prefer to set them unless I have a reason *not* to.
- Browsers are increasingly enforcing `HttpOnly` and `Secure`, e.g., `https://webkit.org/blog/8613/intelligent-tracking-prevention-2-1/`.

**Expected Output:**

```
1 {
2   "solution_leakage_detected": true,
3   "reason": "The solution is explicitly provided in the description.",
4   "extracted_solution": "LANGUAGE_COOKIE_SECURE = False, LANGUAGE_COOKIE_HTTPONLY = False, LANGUAGE_COOKIE_SAMESITE = None"
5 }
```

EXAMPLE 2

**Description:** Shape of `coef_` is incorrect for `linear_model.Lasso` when using `fit_intercept=False`.
**Steps to Reproduce:**

```python
import numpy as np
from sklearn import linear_model

est_intercept = linear_model.Lasso(fit_intercept=True)
est_intercept.fit(np.c_[np.ones(3)], np.ones(3))
assert est_intercept.coef_.shape == (1,)

est_no_intercept = linear_model.Lasso(fit_intercept=False)
est_no_intercept.fit(np.c_[np.ones(3)], np.ones(3))
assert est_no_intercept.coef_.shape == (1,)
```

**Expected Output:**

```json
{
  "solution_leakage_detected": false,
  "reason": "The description identifies a bug but does not explicitly provide a solution.",
  "extracted_solution": null
}
```

——

EXAMPLE 3

**Description:** A typo in `Poly3DCollection.__init__()` causes a `TypeError` exception when calling the function with `shade=True`.
**Relevant Code:**

```
matplotlib/lib/mpl_toolkits/mplot3d/art3d.py

Line 908 in f7a8cab

 if facecolors is None and edgecolors in None:
edgecolors in None should be edgecolors is None
```

**Expected Output:**

```json
{
  "solution_leakage_detected": true,
  "reason": "The solution is explicitly provided as a corrected code snippet.",
  "extracted_solution": "edgecolors in None should be edgecolors is None"
}
```

## A.4 LLM USAGE

We used a large language model (LLM) solely for proofreading and minor copyediting (grammar, clarity, and style). All suggestions were verified by the authors, who remain responsible for the final content.

