# OpenReview forum: "SWE-Bench+: Enhanced Coding Benchmark for LLMs"
_ICLR.cc/2026/Conference — ICLR 2026 Conference Withdrawn Submission_

### Official Review · Reviewer_Md8K · 2025-10-18

**Soundness:** 2
**Presentation:** 2
**Contribution:** 2
**Rating:** 2
**Confidence:** 3

**Summary:**

Point out the lackness of a popular benchmark for software engineer: “solution leakage” and weak test cases. Conduct manually check for the swe-bench.  Create a more rigorous evaluation framework, SWE-Bench+, introduce
SoluLeakDetector, an LLM-based tool to filter issues with solution leaks,
and TestEnhancer, an LLM-based approach to strengthen test suites and mitigate weak test problems.

**Strengths:**

1. Empirical observation of benchmark flaws – The authors perform a manual analysis of SWE-Bench instances and identify two concrete issues: “solution leakage” and weak test cases.
2.  Develop two tools: SoluLeakDetector and TestEnhancer.
3. Quantitative analysis – Provides clear before/after comparisons showing performance drops.

**Weaknesses:**

1. Trivial or incremental contribution – The main idea (“filter leaks and add tests with an LLM”) is conceptually straightforward and mostly engineering.
2. Unclear definition – “Solution leak” is defined informally (direct vs. hint leaks) but lacks precise operational criteria or reproducible quantitative thresholding.
3. Limited distinctiveness of SWE-Bench+ vs. SWE-Bench. Although the paper filters leakage cases and strengthens tests, it doesn’t convincingly show that SWE-Bench+ measures new or different capabilities beyond the original SWE-Bench; most of the reported gains are resolution-rate drops due to stricter tests rather than evidence of qualitatively different skills. There’s no analysis of rank reordering, skill taxonomy shifts, or task distribution changes demonstrating distinct ability coverage.
4. No safeguards against LLM hallucination in both pipeline stages. Both core components are LLM-driven. As a result, hallucinations or non-determinism in either worker could introduce false positives/negatives and shift leaderboards for the wrong reasons.
5. Writing and presentation issues – Some typos (“SoluLeakDetector” capitalized inconsistently, missing commas, unclear figures).

**Questions:**

1. How exactly is solution leakage operationalized? Could the authors formalize a reproducible metric (e.g., token overlap between issue text and gold patch)?

2. What is the ranking difference between SWE-Bench+ and the original SWE-Bench leaderboard? Does it alter model ordering?

3. In real software-engineering settings, issue threads naturally include hints or partial solutions. Why should these be excluded? Wouldn’t understanding and using hints reflect genuine developer assistance rather than “leakage”?

4. Can TestEnhancer generalize beyond SWE-Bench without extensive manual setup?

5. How do you ensure LLM hallucinations from either worker don’t contaminate the benchmark?

---

### Official Review · Reviewer_D1Rh · 2025-10-28

**Soundness:** 2
**Presentation:** 1
**Contribution:** 2
**Rating:** 2
**Confidence:** 5

**Summary:**

The paper examines official submissions to the SWE-bench leaderboard and considers cases in which the issue description describes a valid solution candidate (”solution leakage”) and where the test coverage is too lenient and passes plausible but not altogether correct solutions. The paper then introduces LLM-based approaches for mitigating both of these issues, namely by developing prompts to identify solution leakage and for writing additional tests to enhance robustness, and validating their accuracy on the manually annotated SWE-bench submissions.

**Strengths:**

* The study considers 651 submitted patches where tests passed and conducts a manual audit of these cases. This was conducted by all three authors and disagreements were resolved via discussion.
* The subsequently mitigation strategies are validated appropriately.

**Weaknesses:**

* In my view, it is not very clear why some amount of “solution leakage” is a fundamental problem in the benchmark. While it does make the benchmark easier, it models realistic cases of users approaching models with suggestions.
* Furthermore, the paper’s analysis suffers from a positive evidence bias by only considering passing solutions where the AI-generated patches passed.
* While these defects impact the absolute performance numbers, I am not fully convinced that  make SWE-bench less useful for a relative comparison of different models and approaches. For example, the authors do not show whether any recent models exploit these weaknesses.
* While SoluLeakDetector may increase the hardness of the benchmark, it seems that SWE-bench Verified is saturating regardless and we will need novel and more challenging coding benchmarks instead.
* The paper ignores the possibility of overly specific tests, i.e. false negatives, which are also known to be an issue with SWE-bench.
* It would be good if the authors demonstrate that the identified issues and proposed fixes transfer to the various SWE-bench-style follow ups, such as SWE-bench-multilingual / SWE-bench-Pro / Multi-SWE-bench.

**Questions:**

It would be good to revisit the methodology behind Verified and explain why it failed to unearth the issues, as well as include a broader discussion of related works which also performed critical assessments of SWE-bench:
- Wang et al., Are "Solved Issues" in SWE-bench Really Solved Correctly? An Empirical Study
- Yu et al., UTBoost: Rigorous Evaluation of Coding Agents on SWE-Bench, ACL 2025
- Liang et al., The SWE-Bench Illusion: When State-of-the-Art LLMs Remember Instead of Reason

---

### Official Review · Reviewer_exjW · 2025-10-30

**Soundness:** 3
**Presentation:** 1
**Contribution:** 2
**Rating:** 4
**Confidence:** 4

**Summary:**

This paper presents an empirical analysis of the popular SWE-bench software engineering benchmark, revealing critical underlying issues. The main problems identified are:
1. Solution Leakage: The issue context, such as the bug report or comments, often contains information that implicitly aids in fixing the bug.
2. Weak Unit Tests: The original benchmark's tests are insufficient to cover all edge cases, leading to incorrect solution patches being erroneously accepted as correct.

After filtering these problematic instances, the performance of common software engineering agents dropped significantly.

To address these shortcomings, the paper introduces two LLM-based tools: SoluLeakDetector and TestEnhancer. SoluLeakDetector uses a GPT-4 prompt to classify whether the bug context contains relevant solution information. TestEnhancer provides the LLM with coverage information from the original tests, enabling it to generate more robust and helpful unit tests.

**Strengths:**

This paper offers a highly significant empirical analysis of the SWE-bench benchmark, a critical standard for evaluating the software engineering capabilities of Large Language Models.
The study's key contributions, include:
1. The paper systematically identifies two major flaws undermining the benchmark's reliability: Solution Leakage and Weak Unit Tests.
2. Solution Leakage is further categorized into direct leakage and indirect leakage, providing a nuanced understanding of how bug-fixing information is inadvertently exposed. And Weak Unit Tests are classified based on the nature of the accepted, yet flawed, patches: incorrect fixes, changes made to different files/functions, and incomplete fixes.
3. The research provides intuitive examples for many of these problematic instances, effectively clarifying the detrimental impact of these issues on the benchmark's integrity.
4. The technique of utilizing test coverage information for test enhancement is a reasonable and valuable contribution. This approach holds significant promise and could be broadly adopted across various test generation methods to improve test set rigor.

**Weaknesses:**

1. The overall writing quality of the paper is somewhat coarse. Notably, there is even a reference in the abstract that cannot be properly redirected. The description of the experimental setup is also vague. For instance, Figure 4 is likely to confuse readers regarding the meaning of “overall,” and its caption appears overly simplistic and underdeveloped.
2. The overall contribution of the paper is rather limited. It primarily focuses on identifying and correcting issues within SWE-bench without uncovering deeper underlying problems or proposing more generalizable solutions. For example, the issues discussed—such as data filtering on GitHub and the non-standard or incomplete nature of test cases—represent fundamental challenges. Addressing these problems directly would be more valuable than merely refining an existing dataset.
3. The paper lacks methodological innovation and relies heavily on manual effort. Moreover, the classification of SWE-bench issues appears questionable. For instance, in the category “Different files/functions changed,” many bugs could reasonably have multiple valid fixes or be located in different positions. It is therefore not rigorous to treat this category as an inherent weakness.

**Questions:**

Refer to Weakness

---

### Official Review · Reviewer_82Zt · 2025-11-01

**Soundness:** 2
**Presentation:** 3
**Contribution:** 2
**Rating:** 4
**Confidence:** 4

**Summary:**

In this paper, the authors provides a detailed empirical study on the SWE-bench Verified and SWE-bench Lite and identified two major quality issues existed: solution leak and weak tests. The authors further propose two LLM-based tools: SoluLeakDetector and TestEnhancer to address these two issues and propose a new benchmark SWE-bench+. Experiments on the major methods experience a significant performance drop on SWE-bench+.

**Strengths:**

1. The authors provide a complete quality analysis based on manual check and empirical study. Moreover, the authors provide high-level insights into the solution leakage and weak test issue, as well as fine-grained example based analysis.

2. The authors provide quantitative identification of critical flaws in the existing benchmarks with sufficient statistics.

3. The authors design two automatic LLM tools - SoLuLeakDetector and TestEnhancer, which are well-motivated and have potential use in other code generation benchmarks.

**Weaknesses:**

1. Although the authors present an in-depth manual analysis on SWE-bench Verified and SWE-bench Lite, it remains unclear how the issue of solution leakage is fairly quantified. In real-world problem-solving scenarios, some issues inherently include partial hints as part of clarifying the task instructions, which may complicate defining and measuring leakage consistently.

2. Given the growing concerns about data leakage in LLMs, many benchmarks evolve and update frequently. However, the authors’ analysis relies heavily on the SWE-bench series and therefore exhibits limited generalization.

3. The proposed SoluLeakDetector achieves only around 80% accuracy, indicating limited capability in fully addressing the leakage problem. However, the authors don't provide a clear justification for why this level of accuracy is considered acceptable.

4. There is also no rigorous evaluation for TestEnhancer. The authors mentioned that test cases in SWE-bench Verified and Lite do not guarantee complete coverage and often miss edge cases, but they fail to provide sufficient evidence that TestEnhancer effectively mitigates these limitations rather than only solving it partially.

**Questions:**

1. How exactly can the proposed framework be extend to other benchmarks beyond SWE-Bench? Will it cost too much human efforts? It would be better if the author can further explain this point.

2. How do the authors ensure the quality and correctness of the new test suites generated by TestEnhancer? Is there a quantitative assessment to ensure the reliability of these enhanced tests?

---

### Note · Authors · 2025-12-01

I have read and agree with the venue's withdrawal policy on behalf of myself and my co-authors.